# Impact of Sex and Gender on Clinical Management of Patients with Advanced Chronic Liver Disease and Type 2 Diabetes

**DOI:** 10.3390/jpm13030558

**Published:** 2023-03-20

**Authors:** Anna Licata, Giuseppina T. Russo, Annalisa Giandalia, Marcella Cammilleri, Clelia Asero, Irene Cacciola

**Affiliations:** 1Internal Medicine & Hepatology Unit, University Hospital of Palermo, PROMISE, University of Palermo, 90127 Palermo, Italy; 2Internal Medicine and Diabetology Unit, University of Messina, 98125 Messina, Italy; 3Internal Medicine and Hepatology Unit, University Hospital of Messina, 98124 Messina, Italy; 4Department of Clinical and Experimental Medicine, University of Messina, 98124 Messina, Italy

**Keywords:** cirrhosis, type 2 diabetes, gender, sex

## Abstract

Gender differences in the epidemiology, pathophysiological mechanisms and clinical features in chronic liver diseases that may be associated with type 2 diabetes (T2D) have been increasingly reported in recent years. This sexual dimorphism is due to a complex interaction between sex- and gender-related factors, including biological, hormonal, psychological and socio-cultural variables. However, the impact of sex and gender on the management of T2D subjects with liver disease is still unclear. In this regard, sex-related differences deserve careful consideration in pharmacology, aimed at improving drug safety and optimising medical therapy, both in men and women with T2D; moreover, low adherence to and persistence of long-term drug treatment is more common among women. A better understanding of sex- and gender-related differences in this field would provide an opportunity for a tailored diagnostic and therapeutic approach to the management of T2D subjects with chronic liver disease. In this narrative review, we summarized available data on sex- and gender-related differences in chronic liver disease, including metabolic, autoimmune, alcoholic and virus-related forms and their potential evolution towards cirrhosis and/or hepatocarcinoma in T2D subjects, to support their appropriate and personalized clinical management.

## 1. Introduction

In recent decades, the role played by sex and gender difference as a determinant of health has been increasingly recognized by the entire scientific community. In fact, sex and gender may be identified as critical factors in the medical field to the point that they have become modifiers of mortality due to cardiovascular events. Genetic, epigenetic and hormonal influences of biological sex affect the physiology of the disease; similarly, the social constructs of gender influence community, physician and patient behaviour in the healthcare system and interact with pathobiology. Consequently, nowadays we need to consider sex and gender in the approach to the diagnosis, prevention and treatment of diseases as a necessary and fundamental step towards precision medicine for the health of men and women [1,2].

Chronic liver diseases and type 2 diabetes (T2D) are frequent “partners-in-crime”, because of their large prevalence, common risk factors and the potential risks associated with their coexistence, as the presence of one condition increases the risk of developing deleterious sequelae of the other one. Notably, sex- and gender-related differences in the epidemiology, pathophysiology and clinical aspects have been also described for T2D associated with CLD, and their full recognition is an important step towards the personalization of their management. [3]

In this context, the focus of our attention may be turned to two points: (1) there is a close association between T2D and chronic liver disease, in particular with non-alcoholic fatty liver disease (NAFLD), which currently represents the most common cause of chronic liver disease worldwide, (2) the clinical evolution and response to treatment of both T2D and chronic liver diseases are influenced by several variables, including hormonal factors and lifestyle variables, which may differ between men and women.

However, despite this available evidence, the diffusion of knowledge on sex- and gender-related differences in T2D and chronic liver diseases among health professionals and their clinical application remains limited.

The purpose of this narrative review is to discuss the available evidence on sex- and gender-related differences in chronic liver disease, including metabolic-, autoimmune-, alcoholic- and virus-related forms and their potential evolution towards cirrhosis and/or hepatocarcinoma in T2D subjects, to support their appropriate and personalized clinical management.

## 2. Sex- and Gender-Related Differences in T2D

Worldwide, T2D is more prevalent in men than in women, especially in middle-aged subjects. The recent IDF (International Diabetes Federation) global estimates showed a diabetes prevalence in adults of 10.8% in men vs. 10.2% in women (aged 20–79 years); in 2021, there were 17.7 million more men with diabetes than women [4].

This dissimilarity between the sexes in diabetes epidemiology is partly determined by hormonal and anatomical factors (sex-related), as well as by gender-specific lifestyle habits and behaviours. Relevantly, men are diagnosed with T2D, not just 3–4 years earlier, but also at lower mean body mass index (BMI) values, than women [5]. Moreover, adipose tissue distribution is different between the two sexes, with non-diabetic men generally showing more visceral and hepatic fat, compared to women. Among T2D subjects women are overall more obese, with abdominal obesity present in 70% of women and 40% of men, indicating a different impact of obesity phenotypes on CVD risk [6]. The mechanisms responsible for the higher impact of T2D on CVD risk in women have not been fully clarified yet, but they certainly include the pathophysiology of the vascular bed, and hormonal and biological sex differences. Hence, it can reasonably be sustained that a complex interaction between sex- and gender-related variables is responsible for the greater relative risk of CV complications among women with T2D.

Gender factors include education, social and work roles, as well as access to diabetes centres, therapeutic adherence, quality of life and psychological factors, without neglecting lifestyle, i.e., dietary patterns and physical activity. Sex-related differences in the healthcare provided for the management of diabetes and its complications may contribute to women’s greater relative risk of diabetes complications. Notably, the erroneous perception that women have a lower CV risk than men may cause a delay in the diagnosis and under-treatment of major CVD risk factors: for instance, T2DM women are less frequently prescribed protective drugs, such as statins, and revascularization procedures [3,6]. Moreover, several studies have indicated that middle-aged men are more insulin-resistant than women, generally showing lower HDL cholesterol levels, higher fasting glucose (FPG) and higher triglyceride levels than women of a similar age, even after adjusting for BMI; men also present higher insulin and lower adiponectin and leptin levels, in keeping with different adiposity storage and insulin sensitivity [7,8].

Oestrogen plays a key role in energetic metabolism in pre-menopausal women and post-menopausal oestrogen deficiency contributes to insulin resistance and dysglycemia; on the other hand, testosterone deficiency is partly responsible for insulin resistance in men. In addition, some specific aspects of glucose homeostasis show sex-related differences, with impaired fasting glucose being more prevalent in men, while impaired glucose tolerance is generally more prevalent in women [8,9]. Besides these pathophysiological differences, many other aspects of T2D, such as the clinical presentation and progression of the disease, as well as its chronic complications, are influenced by sexual dimorphism [2,3,6].

Current guidelines recommend the holistic management of all risk factors, including glucose, lipid, body weight and blood pressure levels to prevent T2D complications. Treatment targets for most cardiovascular disease (CVD) risk factors are achieved in different ways in both two genders, albeit more disadvantageously for women with T2D [10,11].

However, the greater relative risk for CVD in T2D women is the most relevant difference that emerges in this field [3,6,7]: women with T2D do not take advantage of the estrogenic protection before menopause, presenting a higher CVD risk even at a younger age [3,11]. Undoubtedly, CVD represents the main cause of morbidity and mortality in subjects with T2D; the overall risk of CVD increases with age, and it appears 15 years earlier than in patients without diabetes. Women with T2D are mostly affected by the vascular complications of diabetes, presenting a 44% higher risk of developing coronary heart disease (CHD) than men [12,13]. The higher relative risk in women also concerns stroke and vascular dementia [14,15]. Women also show a higher in-hospital mortality risk after events, and a higher CVD-related and all-cause mortality, compared with men [16].

Significant gender-related differences in epidemiology, aetiology and associated clinical variables have also been described for heart failure (HF), with some authors demonstrating a greater risk for preserved ejection fraction-HF in women with T2D [17]. The available evidence on potential sex-related differences in peripheral artery disease is more complex to interpret, with some studies suggesting a significantly higher relative risk in T2D women, and others showing a higher risk of lower limb amputations in men than in women [18,19].

Increasing literature data are available for diabetic kidney diseases and their phenotypes [20], while studies on retinopathy and neuropathy are still insufficient to drive any conclusion.

It is important to consider that, as reported in a large population-based study conducted on 415,294 subjects in Italy, the quality of diabetes care is poorer for women than men with T2D managed by specialists [14]. Sex- and gender-related differences have been also reported in the responsiveness to and side effects of hypoglycaemic drugs, as well as in the adherence to the non-pharmacological approach, as reported in the following paragraphs [21,22,23,24,25].

## 3. Sex and Gender Differences in Chronic Liver Diseases and T2D

Liver disease is a major health threat across Europe, as it represents the leading cause of lost years of working life, after CVD [26]. Although, in the last decade, the availability of drugs able to “cure” hepatitis C virus (HCV) and to control hepatitis B virus (HBV) infection has reduced the causes of liver decompensation [27,28], the epidemiological increase of alcohol abuse and metabolic diseases, including T2D and obesity, are changing the epidemiological face of liver disease [29]. Sex- and gender-related differences have been reported in chronic liver diseases of different aetiologies, including alcoholic and non-alcoholic fatty liver disease, chronic hepatitis B and C virus infection, autoimmune-related liver damage, as well as their most severe forms such as cirrhosis and hepatocarcinoma, as reported in the following sections.

Currently, the evaluation of ALT levels is the most used surrogate marker of liver injury in clinical practice. A slight–moderate increase in ALT levels may mark chronic liver diseases due to a large spectrum of aetiologic factors including hepatitis B and/or C viral infections, alcohol intake, T2D, obesity, etc. [30,31]. Despite the impact of sex- and gender-related variables not always being considered, recent data has shown that in subjects with T2D, the alterations in liver enzymes appear differently in the two sexes [32]. Moreover, it appears important to underline that liver disease outcomes appear to be significantly influenced by different access to health services between two genders [33]: in fact, unequal access to care negatively affects the prevalence and the management of liver disease causing a delay in its diagnosis [34].

### 3.1. Alcoholic Liver Disease

Increased alcohol abuse has been recorded in the general population in recent decades; however, the effect of this behaviour appears to be different between the sexes [30]. Significantly, the increase in the rate of alcoholic liver disease (ALD) in the female sex represents a highly important observation [35,36] as also reported in the National Health and Nutrition Examination Survey (NHANES) study showing that between 2015 and 2016 the prevalence of ALD in women increased from 0.5 to 1.3% [37]. The cause of this increase of ALD in females seems to be linked in part to their greater biological susceptibility to the effects of alcohol, defined as a lower volume of distribution and a lower concentration of alcohol dehydrogenase than in males [38,39]. In addition, social disparities within access to treatment (gender issues) play a significant role in determining a worse evolution of ALD in females [40]. In fact, women are both more likely to have partners, friends and family members supportive of their alcohol abuse, and less likely than men to receive drugs used in the prevention or relapse of alcohol abuse such as disulfiram, naltrexone and acamprosate (HR, 0.85; *p* < 0.001) [40].

A further worsening of the prevalence of ALD in women seems to be linked to the COVID-19 pandemic; in fact, recent data show that hospital admissions for alcohol-associated hepatitis have dramatically increased for women [41,42].

Several studies have demonstrated the association between T2D and alcohol consumption. It has been demonstrated that moderate alcohol consumption may have a protective role against insulin resistance, potentially improving insulin sensitivity and glycated haemoglobin values [43]. However, the risk of developing T2D in “heavy drinkers” (alcohol intake > 50 g/day) may be stronger in females than males [44]. In addition, both alcohol intake and obesity represent risk factors for the development of T2D and the onset of macrovascular disease [45]. Moreover, the risk of diabetes appears to be related to the duration of alcohol consumption and it decreases after abstinence [46]. Sex- and gender-related differences and risk factors for ALD are reported in Table 1.

### 3.2. Non-Alcoholic Fatty Liver Disease

Non-alcoholic fatty liver disease (NAFLD) is the most frequent form of chronic liver disease in Western countries [27,56] and the second leading cause of liver transplantation after HCV-related cirrhosis [57,58]. It encompasses a broad spectrum of conditions ranging from fatty liver disease to non-alcoholic steatohepatitis (NASH), cirrhosis and hepatocellular carcinoma, especially when it occurs in T2DM subjects [59,60].

The relationship between NAFLD and T2D is so close and complex that it is difficult to define which of the two conditions may influence the onset and evolution of the other. In fact, if, on one hand, NAFLD could be considered a long-term complication of T2D, on the other NAFLD exacerbates hepatic insulin resistance and increases the risk of developing T2D, worsens glycaemic control and contributes to the pathogenesis of long-term complications, including cardiovascular disease (CVD) and chronic kidney disease [59,61]. Epidemiological data show that the overall prevalence of NAFLD ranges from approximately 20% in the general population to 55–65% among diabetic patients, with an important difference between the two sexes based on age groups. In fact, although men show an overall higher risk of NAFLD, women aged >50 years have a higher risk of NASH and advanced fibrosis, which is associated with an increased risk of mortality [53]. Therefore, it may be suggested that a key role in the onset of NAFLD is played by hormonal and biological variables, including changes in body composition, increased BMI and intra-abdominal fat, with a consequent higher risk of T2D and insulin resistance. Oestrogens protect against fat accumulation and the development of fibrosis in the liver, thus protecting pre-menopausal women against NASH [54,55]. Sex- and gender-related differences and risk factors in NAFLD are reported in Table 1. Furthermore, the worse metabolic profile of the diabetic woman over 60 years of age, characterized by changes in BMI, waist circumference, HOMA-IR and hsCRP, and lipid profile is associated with an increased cardiovascular risk in women as compared with that of men [62].

### 3.3. Chronic Viral Infection: Hepatitis B and Hepatitis C

Data from the World Health Organization (WHO) show that, to date, 257 million and 71 million people have been affected by the hepatitis B virus (HBV) and hepatitis C virus (HCV), respectively [28,63]. The high effectiveness of direct-acting antiviral drugs (DAAs) which allows HCV to be eliminated in over 98% of cases, and the use of nucleoside analogues which are able to block HBV replication, have changed the clinical course of virus-related liver disease. Moreover, a clinical benefit of DAA has also been reported for metabolic outcomes in T2D patients [64].

Data on potential sex-related differences in the response to DAA are still sparse and conflicting. Thus, sex-related differences with regard to the risk of acute cardiovascular complications were analysed in a large e US-based multisite cohort of HCV patients, with observation starting at the time of HCV diagnosis (untreated) or when the last antiviral treatment started (treated). After controlling for other risk factors, sustained virological response (interferon-based (IFN) or direct-acting antiviral (DAA)) significantly reduced the risk of all acute cardiovascular outcomes, particularly among female patients [65]. DAA treatment has also been associated with a smaller reduction in all-cause mortality for females (aHR, 0.50 (95% CI, 0.30–0.85)) compared with males (aHR, 0.34 (95% CI, 0.25–0.46)) [66].

A recent study using neural network machine learning algorithms identified sex among the principal predictors of response to DAA treatment [67]. Conversely, other authors have reported that sustained virological response did not vary based on sex [68].

“Sex-related” hormonal and immunological factors may play a role in the spontaneous clearance of viruses and the outcome of liver disease. In fact, epidemiological data show that chronic HBV infection is more frequent in men than in women (10.7% vs. 4.4%) [69]. Although the molecular mechanisms are not fully understood, oestrogens and androgens have opposite effects on the regulation of HBV DNA transcription [47]. A key role is certainly played by the presence in the liver of oestrogen receptors, which regulate the immune response against HBV [48]. Furthermore, androgens promote HBV transcription while oestrogens suppress B virus mRNA levels, with opposite effects on infection [47].

Even the progression of liver disease seems to be different in the two sexes, and several clinical studies have shown that HBV and HCV chronic hepatitis evolve more rapidly towards cirrhosis in men and in postmenopausal women [48]. Furthermore, male gender, along with age and alcohol intake, are important risk factors for the progression of HCV-related liver disease [70].

A relevant role in determining the evolution of the virus-related liver disease is certainly also played by metabolic factors, including T2D [71]. HCV is a risk factor for insulin resistance, metabolic syndrome and type 2 diabetes mellitus [72,73]. Epidemiological data show that the prevalence of T2DM in patients with chronic HCV infection is estimated to be between 30% and 70% [74]. HCV causes the inhibition of sodium–glucose co-transporters (SGLTs) in the liver, leading to a deficiency in glucose uptake in hepatocytes and of the kinase that regulates insulin sensitivity through the action of HCV non-structural proteins (core, E1, E2, and MS3-NS5B); it also enhances the inflammatory response by the release of tumour necrosis factor α (TNF-α) and interleukine-6 (IL-6) [75]. On the contrary, HBV does not appear to be related to glycolipid impairment [76,77]. Epidemiological data has shown that in HBV-positive patients the prevalence of insulin resistance and metabolic syndrome is lower than in the general population. Sex- and gender-related differences and risk factors in chronic HCV- and HBV-related hepatitis are reported in Table 1.

### 3.4. Autoimmune Hepatitis

In the general population, autoimmune diseases are more frequent in females than in males, with a prevalence of 80% for women [49]. Although several factors such as sex hormones, and genetic and epigenetic factors seem to be involved in the association between sex and autoimmune disease, many aspects are still to be clarified. Autoimmune hepatitis (AIH) confirms the trend of a higher prevalence in the female gender [50,51], both in the paediatric and adult populations, with a male/female ratio of 1:4 in the AIH type 1 and 1:10 in the AIH type 2 [49,50,51,52]. Females are also more prone to overlap AIH/CBP syndromes than males [51]. Clinical data show a close correlation in the female gender between AIH and diabetes. Furthermore, the pro-inflammatory state that characterizes autoimmune diseases can influence glucose metabolism, causing insulin resistance. In this context, the treatment of T2D appears more complex, due to the simultaneous use of corticosteroid drugs which make it more difficult to achieve glycaemic compensation and require higher doses of insulin or oral hypoglycaemic drugs [78]. Sex- and gender-related differences and risk factors for autoimmune hepatitis are reported in Table 1.

It may be concluded that the simultaneous presence of T2D and autoimmune conditions worsen the outcome of both pathologies. In women with autoimmune diseases and diabetes, there is in fact both a higher incidence of micro- and macrovascular complications and a worsening of advanced liver disease to hepatocellular carcinoma.

### 3.5. Hepatocellular Carcinoma

Hepatocellular carcinoma (HCC) represents one of the most fatal human neoplasms. It generally arises on a cirrhotic liver and, regardless of the aetiology of the cirrhosis itself, it is more frequent in men than in women [79]. Epidemiological data show that HCC is the fifth most common malignancy in men and the eighth most common in women worldwide, with male/female ratios averaging between 2:1 and 4:1 [80]. Currently, both the prevalence and mortality for HCC in both sexes are increasing worldwide [81].

In patients with NAFLD the prevalence of HCC is 2.6%, four times higher in males than females [54,82]. Although several factors are decisive in the onset of HCC, such as advanced age, pro-inflammatory cytokines, the action of adipokines, smoking habits, the presence of obesity and T2D, a key role is certainly played by sexual hormones [83,84]. Oestrogens play a protective role against liver disease, but, as demonstrated by the epidemiological trend of liver cirrhosis and HCC, this protection is drastically reduced in women aged >65 years [85].

The clinical presentation and outcome of HCC may also differ in men and women [85]. Overall, in female patients, the progression of HCC is slower than in males, with better survival [85,86]. Male patients have higher values of α-fetoprotein, bilirubin and Mayo end-stage liver disease (MELD) scores than females. The size of the HCC and the rate of vascular invasion and extrahepatic spread are also frequently greater in men [53]. By contrast, women with HCC have a more conserved residual liver function, characterized by higher levels of albumin and platelets and lower levels of INR and MELD score—a picture associated with a good response to liver resection [53].

All these sex-related differences in the development and clinical course of HCC should be taken into account in order to carry out early diagnosis and to guide tailored surveillance strategies [60]. However, it is important to underline that HCC patients with diabetes, obesity and cardiovascular disease, despite a potential early diagnosis, are likely to receive a more adverse prognosis, as the presence of such comorbidities may limit access to treatments such as resection and liver transplantation.

## 4. Sex- and Gender-Related Differences in the Gut Microbiota of Patients with NAFLD and T2D

NAFLD and NASH are more prevalent among males and are associated with obesity, dyslipidemia, T2D and CVD. These conditions are reported to be different between the sexes. Increased gut permeability and bacterial translocation within portal circulation have been reported in NAFLD, indicating that the gut–liver axis represents a source of systemic and hepatic inflammation and reflects the activity of the gut microbiome [87]. Several studies have demonstrated that endotoxemia reflects an increase in intestinal permeability and is involved in the progression of liver damage [88]. Interestingly, the gut microbiome may display sexual dimorphism as a consequence of genotype, diet, age, ethnicity, geographic location and/or the health status of the host.

Evidence from studies suggests that women may harbour a higher ratio of Firmicutes/Bacteroidetes (F/B) in comparison to men [89]. The F/B ratio, increasing in magnitude from birth to adulthood, is used in microbiome studies as it is an important measure of human microbiota composition and appears to be a key component in biological ageing and obesity. Among those with a BMI greater than 33, a significantly lower F/B ratio has been seen in men compared to women, while the opposite holds true in those with a BMI lower than 33 as well as in postmenopausal women [90]. Adjusting for BMI, higher proportions of Firmicutes have been found in women compared to men. With respect to other less represented gut microbiome phyla, higher numbers of Proteobacteria, Veillonella and Blautia have been reported in women in comparison to men. The F/B ratio has been used as an indicator of gut dysbiosis, with a higher F/B ratio representing a more dysbiotic microbiome. In addition to compositional differences, sex-specific heterogeneity may exist in microbiome responses to external stimuli, including diet. Dysbiosis has been associated with many diseases, including T2D, cardiovascular disease, diastolic dysfunction and LVH, by provoking variations in the composition of the gut microbiota (GM), altering the intestinal wall permeability and increasing the secretion of metabolic endotoxins [88,91,92]. However, the proportion of Firmicutes is significantly lower in diabetic patients compared to healthy individuals, and the F/B ratio has been significantly associated with impaired glucose metabolism [93]. Other studies have suggested that dysbiosis may play a role in cardiovascular disease [94,95]. Recently, a study [96] exploring the associations between GM and subclinical features of cardiovascular disease in diabetic patients showed that a high concentration of phylum Bacteroides was significantly correlated with a study [96] exploring the associations between GM and subclinical features of cardiovascular disease in diabetic patients showed that a reduced concentration of phylum Bacteroides was significantly correlated with left ventricle systolic and diastolic dysfunction. left ventricle systolic and diastolic dysfunction. Moreover, reduced concentration of phylum Firmicutes was associated with a higher risk of left ventricular hypertrophy. Finally, a high phyla F/B ratio and high concentration of the genera Bacteroides were significantly correlated with left atrium enlargement. Hence, it may be summarized that alterations in the gut microbial composition might be used for predicting the development of subclinical cardiovascular disease in diabetic patients.

It has been shown that dysbiosis can be resolved with dietary supplements, such as prebiotics, polyphenols, flavonoids and bioactive compounds, as the latter have antioxidant and antimicrobial effects on the GM [97]. Modulation of GM by bioactive compounds helps the prevention of diabetes by improving glucose homeostasis and insulin sensitivity and by increasing the production of short-chain fatty acids. Further studies are needed to clarify the complex interplay between dietary factors and the combination of bioactive compounds, as well as their action on GM, and the interactions between GM, age, gender and their impact on the onset, prevention and management of type 2 diabetes (Figure 1).

## 5. Sex- and Gender-Related Differences in the Management of Chronic Liver Diseases and T2D

### 5.1. Non-Pharmacological Therapy

Considering the available literature data, it may be assumed that women have healthier lifestyles and diets than men, especially with regard to meat, poultry, fruit and vegetable consumption [98]; it is well established that red processed meat, high-fat foods and fructose are all risk factors for NAFLD development, while high consumption of vegetables and fresh fruit has a protective role (Mediterranean diet). Beyond the quantity of eaten food, we may have also to consider meal timing, because skipping meals or late-night dinners may contribute to higher risk of developing hepatic fibrosis. All these data should be contextualised with the socio-economic status of the considered population. Another element that should be considered when we talk about lifestyle differences in genders is drinking habits [98]: according to Dallas Heart Study, NAFLD prevalence is superior in male more than female patients (42% vs. 20%) with the same alcohol intake.

Current knowledge on the management of post-NASH liver cirrhosis has not, up till now, sufficiently taken into account the aspects related to sex and gender which are increasingly relevant in many chronic diseases, as sex hormones influence lipid metabolism and oxidative stress within the liver and adipose tissue [98,99].

Lifestyle changes appear to lead to changes in DNA methylation [100]. It was already known that reduced levels of global DNA methylation can be associated with genomic instability and thus may become independent predictors of cancer risk. All these reasons make lifestyle changes essential. It should also be mentioned that the global methylation of leukocyte DNA differs by sex and race/ethnicity, suggesting the importance of taking these variables into consideration [101]. Treatment options for NAFLD and NASH include a wide range of options from lifestyle modification (diet and physical activity) that appear to improve portal hypertension [102] to pharmaceutical interventions, bariatric surgery and finally, liver transplantation. Therefore, the role of overnutrition and insulin resistance in the pathogenesis of steatosis and NAFLD, resulting in an imbalance between the accumulation and disposal of triglycerides within the liver, should be acknowledged.

A first step to impede the aforementioned accumulation would necessarily consist of changing eating and physical activity habits, as well as promoting weight reduction. In this regard, it appears that fructose has a negative impact on liver fibrosis and, therefore, should be avoided, while the consumption of coffee appears to be protective [103].

Physical activity improves insulin sensitivity and fatty liver disease; cardiorespiratory fitness is recommended as an activity that can reduce liver damage. These recommendations do not apply in the case of patients with advanced cirrhosis and sarcopenia, cachexia and/or protein deficiency [104]. A detailed description of the pharmacological approach to advanced chronic liver disease and T2D is beyond the scope of this narrative review; however, the following paragraphs summarize the main lipid-lowering and hypoglycaemic drugs available for the treatment of T2D and NAFLD, focusing on sex- and gender-related differences when literature data are available.

Sex- and gender-related differences in the management of T2D and chronic liver disease (CDL) are reported in Table 2.

### 5.2. Statins

Statins, after a long and controversial debate, are now becoming the most commonly used drugs in liver disease of metabolic origin. Cholesterol, which is capable of activating hepatic stellate cells, determines an exacerbation of hepatic fibrosis, as has been observed in experiments on mice, and in liver carcinogenesis [127]. Statins act both by inhibiting β-Hydroxy β-methylglutaryl-CoA (HMG-CoA) reductase and inhibiting Rho kinase, and by stimulating the intrahepatic production of nitric oxide, with the combined effect of reducing intrahepatic resistance [128], improving liver function tests and possibly delaying liver decompensation. Statins are also safe in cirrhotic patients [105], although they remain contraindicated in the datasheet of the drug package. A recent meta-analysis showed that the use of statins also reduced the risk of hepatocellular carcinoma [106]. Therefore, chemoprevention against HCC could be considered useful with drugs to control cholesterol levels [129]. Furthermore, patients with cirrhosis on statin treatment experience fewer infections than others [130]. Despite the overwhelming evidence of the benefits of lipid-lowering medications, women are often undertreated in clinical practice [131], and they have been shown to experience more side effects than men, causing a higher rate of discontinuation [132]. In a large T2D cohort, it has been demonstrated that LDL-C management is worst in T2D women, who are monitored and reach targets less frequently than men. This gap is wider in older women with long-standing T2D [133].

### 5.3. Ezetimibe

Several studies evaluating the efficacy of ezetimibe, which is a potent inhibitor of cholesterol absorption, in treating non-alcoholic fatty liver disease (NAFLD) and non-alcoholic steatohepatitis (NASH), have shown inconsistent results. In a recent meta-analysis [107] including six studies (two RCTS and four single-arm trials) and a total of 723 patients, the authors found that ezetimibe significantly reduced serum aspartate aminotransferase, alanine aminotransferase and Ƴ-glutamyl transpeptidase levels, as well as hepatic steatosis and hepatocyte ballooning. However, hepatic inflammation and fibrosis did not improve in patients with NAFLD and NASH. In randomized controlled trials, only hepatocyte ballooning improved with ezetimibe treatment. In the ESSENTIAL study [108], ezetimibe was added to rosuvastatin versus rosuvastatin monotherapy to assess the reduction in liver fat evaluated by using magnetic resonance imaging-derived proton density fat fraction (MRI-PDFF) in patients with NAFLD. This study showed that the combination therapy significantly reduced liver fat compared with MRI-PDFF monotherapy (mean difference: 3.2%; *p* = 0.020). Furthermore, it was reported that the controlled attenuation parameter (CAP) was significantly reduced by transient elastography in the combination group (321 to 287 dB/m; *p* = 0.018), but not in the monotherapy group (323 to 311 dB/m; *p* = 0.104). Thus, ezetimibe and rosuvastatin were found to be safe for treating patients with NAFLD. Male sex seems to be an independent predictor of total cholesterol reduction in T2D subjects treated with ezetimibe [134] whereas the CVD benefits were just confirmed to be similar in both sexes [135].

### 5.4. Fibrates

Fibrates are a widely used class of lipid-lowering drugs for the treatment of hypertriglyceridemia. Through the activation of the nuclear peroxisome proliferator-activated receptors (PPARs), they regulate the beta-oxidation of lipids. However, it has been demonstrated that fibrates cause side effects such as liver dysfunction and increase creatinine levels; moreover, some clinical trials have reported negative results for the prevention of atherosclerotic cardiovascular diseases [136]. This could be explained by their low selectivity and potency for binding to PPAR-α. To overcome these concerns, novel selective PPAR-α modulators (SPPARMα) with a superior benefit–risk balance compared to conventional fibrates have been developed, such as pemafibrate, the first SPPARMα [137]. Clinical studies in Japan have reported its superiority in the reduction of serum triglycerides (TG) and in the increase of HDL cholesterol. Although available fibrates have shown worsening liver and kidney function, pemafibrate showed improvement and did not have a negative effect on the estimated glomerular filtration rate (eGFR). In fact, pemafibrate is metabolized in the liver and excreted in the bile, and thus can be used safely even in patients with impaired renal function. In addition, pemafibrate could be a promising therapeutic agent for NAFLD and NASH as well as a candidate for combination therapy with statins or other lipid-lowering drugs. Numerous studies in the literature have demonstrated that pemafibrate significantly improves liver function, serum TG and liver stiffness in NAFLD patients [109]. Moreover, pemafibrate improves markers of hepatic inflammation and fibrosis regardless of BMI. Subjects with lean NAFLD (BMI < 25) had a greater response to pemafibrate therapy compared to those with obese NAFLD (BMI > 30). For subjects with obese NAFLD, double-dose pemafibrate and/or combined treatment with a sodium–glucose co-transporter 2 (SGLT2) inhibitor should be considered [138]. Finally, the large-scale multicentre randomized trial, Pemafibrate to Reduce Cardiovascular Outcomes by Reducing Triglycerides in Patients with Diabetes (PROMINENT) study for dyslipidemic patients with type 2 diabetes, showed that statin–pemafibrate combination therapy in diabetic patients with dyslipidemia improved lipid metabolism safely, without increasing the risk of liver dysfunction and myopathy. However, the incidence of cardiovascular events was not lower among those who received pemafibrate compared to those who received a placebo [139,140].

Fenofibrate also reduces triglyceride levels, moderately increases HDL cholesterol levels and decreases the synthesis of apolipoproteins, showing anti-inflammatory and antioxidant effects.

Overall, fibrate use is associated with reduced hepatic and extra-hepatic tissue insulin resistance and with the upregulation of the lipoprotein lipase, a key enzyme in the pathogenesis of hepatic steatosis. Although preclinical studies suggested a therapeutic role of fenofibrate in NAFLD, for its antioxidant, antiapoptotic, anti-inflammatory and antifibrotic activity, available human studies are inconclusive. [141]. Some experimental studies have shown a minor protective effect of fenofibrate on CV events in females, partly due to a different expression of PPARs in the two sexes [110], although the implications in humans are not yet known.

### 5.5. Pioglitazone

Pioglitazone, a peroxisome proliferator-activated receptor (PPAR) agonist, is a glucose-lowering agent that reduces insulin resistance in the liver, muscle and adipose tissue and increases lipid storage in the subcutaneous adipose tissue, also improving the serum lipid profile.

In several studies, pioglitazone was associated with significant histologic benefits in terms of liver fat content, NAFLD activity score, and resolution of inflammation and fibrosis, in T2D subjects with NASH [142].

Due to its efficacy for NASH resolution and improvement of advanced fibrosis, current guidelines indicate pioglitazone for T2D patients with biopsy-proven NASH [111]. However, its effectiveness is highly variable, depending on the study populations, and some evidence suggests a greater metabolic benefit of pioglitazone in female patients. In particular, PPAR agonists were shown to be more effective in glycaemic control, lipid improvement and markers of insulin resistance (HOMA-IR) in women than in men [112]. The influence of sex hormones on PPAR-Ƴ expression and function and the different distribution of adipose tissue in the two sexes may partly explain the observed sex-specific different effects of pioglitazone. Moreover, some adverse effects of pioglitazone, such as oedema, occur more frequently in women than in men. However, sex-related differences in drug efficacy on NAFLD deserve further studies. Notably, pioglitazone-treatment-associated risks, including bone fractures, weight gain, heart failure and bladder cancer, may limit its use in daily clinical practice, deserving an appropriate evaluation of the risk–benefit profile in any patient.

### 5.6. Proprotein Convertase Subtilisin/Kexin Type 9 (PCSK9) Inhibitors

There are inconsistent findings regarding the effect of lipid-lowering agents on non-alcoholic fatty liver disease (NAFLD). Proprotein convertase subtilisin/kexin type 9 (PCSK9) is an important player in cholesterol homeostasis and intracellular lipogenesis. The current evidence from a landscape of preclinical and clinical studies examining the role of PCSK9 in NAFLD shows controversial results. Preclinical studies have indicated that PCSK9 is associated with NAFLD and NASH progression. In humans, it has been concluded that the severity of hepatic steatosis affects the correlation between circulating PCSK9 and liver fat content, with a possible impact of circulating PCSK9 in the early stages of NAFLD, but not in the late stages [113].

Recently, a study carried out on alcoholic cirrhotic patients [114] has shown that high serum PCSK9 was reduced in patients with liver cirrhosis in comparison to non-cirrhotic patients; that plasma PCSK9 was not correlated with Child–Pugh score, MELD score, bilirubin or aminotransferases, but it was associated with low levels of certain cholesteryl ester and sphingomyelin species.

As for potential sex-related differences, circulating concentrations of PCSK9 have been reported to be significantly higher in women than in men, especially after menopause, with a potential regulatory role played by oestrogens. Moreover, women were more likely to initiate treatment with PCSK9 inhibitors, although sex-related differences in the CVD benefits of these drugs have yet to be elucidated [115]. Sex-related differences in liver benefits of PCSK9i have not been determined.

### 5.7. Metformin

Metformin represents one of the therapeutic cornerstones in the management of patients with liver disease, even in overt liver cirrhosis, since it may be associated with better survival, lower risk of hepatocellular carcinoma and lower hepatic encephalopathy, possibly reducing the risk of infections and episodes of variceal bleeding [143,144,145]. Metformin works by improving insulin resistance, decreasing the hepatic production of glucose by inhibiting gluconeogenesis and by an action on glucose-6-phosphatase, decreasing glycaemic values and also enhancing the effect of insulin on muscular glucose absorption. In fact, it was already known that hyperglycaemia and hyperinsulinemia are implicated in ageing and in the development of cancer, probably due to chronic increases in insulin growth factor 1 (IGF-1). In this sense, the biguanides seem to work. A study conducted in consanguineous mice treated with metformin slightly modified food consumption, whereas the dynamics of body weight and the average life of female mice (4.4%) were not or were only slightly modified [146]. Metformin is one of the most relevant antidiabetic agents used in patients with cirrhosis; due to insulin resistance and impaired glucose tolerance, these patients typically develop T2D [147]. Other beneficial effects consist of better control of diastolic blood pressure and HDL cholesterol levels. It also influences the overall prevalence of metabolic syndrome in women with PCOS, reducing it from 34.3% to 21.4%, in a dose-dependent manner [148].

We can sustain that some molecular responses of metformin are influenced by sex hormones [116], thus leading to prescribing lower doses of metformin to women than men, who also experience greater gastrointestinal side effects. Nonetheless, the incidence of cardiovascular events in metformin-treated women was lower than in men [149]. It should be pointed out that, up to now, from the gender point of view, there are no known differences in the relevant genetic background. This could represent a starting point for future studies. In patients with liver cirrhosis, for many years, there were worries about the safety profile in relation to the risk of lactic acidosis, which however appears to be rare in patients without acute renal dysfunction due to dehydration, vomiting, diarrhoea, etc., or in elderly patients with low glomerular filtration rates [150]. Zhang et al. found that continuing metformin use after the diagnosis of cirrhosis improves survival by reducing the risk of HCC and death or liver transplantation, reducing the risk of hepatic encephalopathy possibly by inhibiting glutaminase activity and improving sensitivity to insulin [144]. However, due to recent conflicting evidence, it seems more reasonable to use lower doses of metformin in cirrhotic patients, with a GFR lower than 30 mL/min/1.73 m2 [117]. Regarding gender differences, a study [118] aimed at analysing different therapeutic targets in patients treated with metformin showed that women compared to men had a longer duration of the disease, and lost a greater percentage of weight, unlike the men who instead underwent a more significant reduction in the target values of HbA1c, very often treated with combined therapy. This evidence indicated that metformin therapy was responsible for better glycaemic targets among women. Hence therapies should be individualized towards gender-specific effects.

Another subject of growing interest is the combination therapy of exenatide and metformin, which appears to be more effective in female than in male patients [119].

Newer actions of metformin have been described at the intestinal level, as well [151]. Metformin intake in humans is associated with a reduction in Bacteroides phyla. An increased microbial prevalence of Akkermansia muciniphila, is associated with better insulin sensitivity, loss of weight and improved liver fat and function in experimental models [152]. In type 2 diabetic patients given metformin, a great amount of Akkermansia in the GM resulting in a moderate loss of weight was recently described [153].

### 5.8. Dipeptidyl Peptidase-4 (DPP-4) Inhibitors

DPP-4 inhibitors stimulate insulin secretion and reduce glucagon release by inhibiting the inactivation of GLP-1 and glucose-dependent insulinotropic polypeptide (GIP) [154]. The rationale for the use of these molecules arises from the fact that in patients with liver cirrhosis, there is an upregulation of DPP-4i expression [155] while at the same time having an effect on sarcopenia, typical of cirrhotic patients, as they mitigate the decline in muscle mass [156]. As a precaution, vildagliptin should be avoided due to the risk of hepatotoxicity. Looking at the evolution of liver disease, DPP-4i are able to reduce the hepatocellular lipid content, resulting in a reduction in hepatic and myocardial steatosis (a positive effect of sitagliptin was seen on liver histology in NASH after one year of treatment), simultaneously improving heart function, blood pressure and body weight, especially in women [120,121]. The beneficial effect of DPP-4i on the fatty liver is probably due to GLP-1R expressed in human hepatocytes. In fact, DPP-4i, independently of glycaemic control, also exert an effect on postprandial lipidaemia by stimulating the endogenous action of GLP-1 and inhibiting the secretion of hepatic triglycerides through the activity of the protein kinase activated by AMP [157]. For this reason, DPP-4 becomes a possible new interesting biomarker of liver damage in diabetic patients, due to the observation that DPP-4 is highly expressed in the liver, and its concentrations are positively correlated with liver function tests, as well as the correlation with the histopathologic degree of NASH and NAFLD [158].

### 5.9. Glucagon-like Peptide-1 Receptor Agonists (GLP-1 RAs)

GLP-1 receptor agonists (exenatide, liraglutide, lixisenatide, semaglutide and dulaglutide) act, on one hand, by stimulating the release of insulin in response to glucose and, on the other, by reducing the release of glucagon from the α-cells of the pancreas, cumulatively protecting the mass of β cells. In addition to this, GLP-1 receptor agonists induce feelings of satiety by slowing gastric emptying [159,160] resulting in weight loss and reduced risk of hypoglycaemic events (since insulin secretion occurs only when there is a glucose trigger), especially in obese patients [161]. As for its mechanism of action, following the ingestion of meals, the production of GLP-1 is evoked by the intestinal cells, which in turn stimulates the production of insulin by the pancreas and contextual inhibition of glucagon secretion [162]. These molecules have been shown to be particularly useful in liver disease and, in fact, the efficacy of liraglutide in improving NASH had already been demonstrated in the Liraglutide Efficacy and Action in Non-alcoholic Steatohepatitis (LEAN) study [122]. It is already known from previous studies that GLP-1 RAs could improve liver damage and lipid metabolism in patients with liver disease by significantly reducing the fat content of the liver [163], even if the liver does not metabolize the GLP-1 RAs and therefore no dose modification is required in cirrhosis, in a liraglutide pharmacokinetic study [164]. However, it has been shown that caution should be used in advanced liver disease, as we have limited data on their safety and efficacy [123].

Exenatide and lixisenatide are excreted by kidneys, and for this reason they should be avoided if the eGFR is <30 mL/min/1.73 m^2^ [165]. Regarding exenatide, this is effective in metabolic disorders, thanks to significant weight loss, regardless of sex. There are other variables that predict different outcomes at one year: lower baseline HbA1c values and shorter disease duration. Males appear to have a more effective response to exenatide therapy, possibly due to their shorter disease duration, as well as less depleted β-cell mass at baseline, as indirectly suggested by the lower rate of males on combined therapy, unlike women who show a greater effect on weight reduction [124].

Another very promising molecule is semaglutide, which in a recent phase 2 study proved effective in resolving NASH, compared to placebo. Although it does not seem to act on the improvement of the stage of fibrosis [166]. Furthermore, semaglutide administered once weekly also reduces the risk of cardiovascular events compared to placebo, regardless of gender, age or baseline CV risk profile [167].

Meanwhile, in all AWARD studies, dulaglutide demonstrated significant improvements in glycaemic control regardless of gender and other variables, such as diabetes duration or baseline HbA1c values. Dulaglutide is also well tolerated, with a safety profile similar to other glucagon-like peptide-1 receptor agonists [168]. In Japan, a report regarding the use of dulaglutide, also analysing gender differences, found that reduction in HbA1c was not influenced by the patient’s sex, while the greater weight loss or less weight gain was confirmed, together with a higher incidence of adverse events, including nausea, in female patients compared to male ones. Nonetheless, the incidence of patients who discontinued dulaglutide early due to adverse events was low (<10%) for both genders and no new safety concerns related to dulaglutide were identified for either gender. Therefore, the risk/benefit ratio for dulaglutide remains unchanged and remains positive for both sexes. In addition, with regard to episodes of total or nocturnal hypoglycaemia, no differences were observed in incidence between both sexes in any treatment group [169].

Conflicting data on gender differences in incretin-treated patients emerged from a study that considered data from two Korean national databases, where females were about twice as likely as males to report adverse events related to the use of GLP-1 RA, with a reporting ratio M/F of 2.34. Adverse events were predominantly gastrointestinal upset and headache (the latter with a hazard ratio of 7.97), and no significant difference was found between the two sexes in episodes of nocturnal hypoglycaemia [170,171]. Regarding the effects on HbA1c, gender did not prove to be a predictor of a more or less significant reduction, unlike what emerged in several previous studies. Further comparison studies are needed to confirm these data. According to existing studies, no conclusive results have yet emerged as to whether gender can influence other parameters, such as waist circumference (WC), blood pressure (BP), lipid profile and incidence of cardiovascular events [172].

### 5.10. Sodium–Glucose Cotransporter 2 Inhibitors (SGLT-2i)

Sodium–glucose co-transporter 2 inhibitors (SGLT2i) reduce hyperglycaemia by promoting the urinary excretion of glucose and not involving insulin secretion [173]. Their action is carried out through the inhibition of the sodium–glucose co-transporter 2 (SGLT2) located in the S1 segment of the proximal tubule of the kidney, which is the main transporter responsible for the renal reabsorption of glucose, leading to the urinary excretion of glucose (glycosuria). In addition to this known action, this class of molecules (dapagliflozin, empagliflozin, canagliflozin, and ertugliflozin) are able to improve aminotransferase levels and the degree of hepatic steatosis in subjects with T2D and NAFLD [174,175]. Another aspect that makes these drugs appropriate in liver diseases, especially in the phase of decompensation, is that they improve ascites and oedema by preventing the reabsorption of sodium and glucose in the kidneys, similar to the action of furosemide [125], counteracting the splanchnic and peripheral vasodilation typical of cirrhosis. The latter decreases the effective circulating volume and activates the renin–angiotensin–aldosterone system (RAAS), ultimately favouring water and saline retention. SGLT-2 inhibitors can act on this RAAS dysfunction, since reducing sodium reabsorption in the proximal tubule increases its release downstream into the *macula densa* and, consequently, inhibits renin and RAAS secretion. All SGLT-2i are well tolerated and metabolised in the liver. Potential adverse effects of SGLT2i are hypotension, acute kidney injury and genitourinary tract infections.

In rodent experimental models, SGLT2i have shown to suppress the development of NAFLD and/or NASH, improving histological fatty liver disease or steatohepatitis in obese mice or rats with T2D [176,177]. A recent randomized study of dapagliflozin also demonstrated improvement of hepatic steatosis in patients with T2D diabetes and NAFLD, with relief of hepatic fibrosis only in patients with significant hepatic fibrosis [178].

The available evidence suggests that different antidiabetic drugs may have different responses in terms of efficacy and safety, from a gender and weight perspective. The lack of studies on a heterogeneous group of men and women in order to study these differences in the therapeutic field does not allow us to draw definitive conclusions.

A meta-analysis of three randomized trials conducted with SGLT-2I versus placebo demonstrated a significant reduction in major adverse cardiac events in men, but not in women, suggesting different results based on gender [126]. Another meta-analysis of seven randomized trials did not suggest large differences in the risk of cardiovascular events in both men and women, compared to placebo. Distinguishing between drugs, the reduction in cardiovascular events with SGLT-2I appears to be significantly lower in women with diabetes than in men, while the same cannot be said with GLP-1RA, which confers a similar reduction in events regardless of gender. Women appear to derive a greater CV benefit with GLP-1RA than with SGLT-2, possibly due to relatively greater weight loss due to reduced fat mass with GLP-1RA compared to SGLT-2i [179]. More research is clearly needed as it remains to be determined whether these results are due to the bias of underrepresentation of women, inadequate statistical power or actually reflecting a true gender difference.

Interpretation biases may be present in several epidemiological, cross-sectional and cohort studies suggesting that a woman with T2D has less well-controlled HbA1c, lipid levels and blood pressure than her male counterparts. [180]. These gender disparities are further compounded by a higher base body mass index (BMI) at the time of diagnosis of diabetes in women, due to the biological effects of the female sex hormone [181].

Analysing the available evidence (EMPA-REG OUTCOME, DECLARE-TIMI 58, VERTIS CV, DAPA-HF, and EMPEROR-Reduced), with the exception of the CANVAS study, SGLT-2 inhibitors do not confer a significantly decreased risk of major adverse cardiovascular events among women. However, they provide significant results in terms of reduced risk of cardiovascular death or hospitalization for heart failure, driven primarily by outcomes observed in the heart failure population with reduced ejection fraction [182].

## 6. Conclusions

Recognition of sex- and gender-related differences in T2D and in chronic liver diseases, their diffusion and the clinical application of the findings among health professionals are still inadequate, despite the growing body of data on this issue.

Sex and gender influence our lives not only through sexual behaviours and social interactions but also in education, job roles, access to cures, therapeutic adherence, quality of life, psychological factors and lifestyle. Nowadays, the epidemiology of liver disease is changing because the availability of new treatments for HBV and HCV infections has reduced hepatic decompensations, while metabolic diseases are rising. In this context, the relationship between NAFLD and T2DM is close and complex, and the two conditions frequently combine to produce worse clinical outcomes in the affected patients. Looking at this complex clinical landscape, this narrative review attempted to analyse not only how sex- and gender-related differences influence the development of T2D and chronic liver diseases, but also their potential influence on the therapeutic approaches to these pathologies.

Available evidence on sex- and gender-related differences in chronic liver diseases from different aetiologies, including metabolic-, autoimmune-, alcoholic- and virus-related forms and their potential evolution towards cirrhosis and/or hepatocarcinoma in T2D subjects, has been summarized in order to support healthcare professionals in the personalized clinical management of these two worrisome pathologies. From a pharmacological point of view, the data available in the literature confirm that statins have become the most used therapy in NAFLD, while metformin has turned out to be one of the cornerstones in the management of T2D patients with liver diseases. Hints for future investigations are represented by incretins, gliflozins and PCSK9i which, despite the lack of studies in cirrhotic patients, seem to reduce CVD death and/or hospitalization for HF in the general population, representing a starting point for further studies in which sex and gender should be adequately represented.

## Figures and Tables

**Figure 1 jpm-13-00558-f001:**
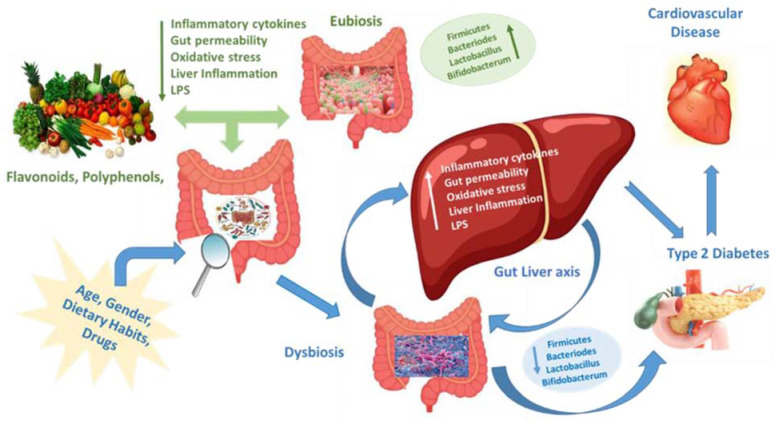
Relationship between food bioactives, gut microbiota and risk factors associated with the development of T2D and cardiovascular disease. Gut eubiosis is maintained by an adequate assumption of polyphenols and flavonoids, commonly contained within fresh fruits, seeds and vegetables, which are the main constituents of the Mediterranean diet. Dysbiosis, influenced by the Western diet, polyunsaturated fatty acid, age, gender and antimicrobials, is characterized by low levels of Firmicutes/Bacteroides and high levels of bacterial lipopolysaccharide, increased gut permeability, high liver inflammation, and high level of oxidative stress and release of pro-inflammatory cytokines. All these changes over a period of time determine the development of obesity, insulin resistance, NAFLD and T2D. In addition, alteration in GM composition associated with T2D raises the risk of left ventricular hypertrophy, left atrium enlargement and systo-diastolic dysfunction.

**Table 1 jpm-13-00558-t001:** Sex- and gender-related differences in risk factors for CLD.

	Male	Female	References
HBV/HCV-related chronic hepatitis	Androgens promote HBV transcription.	Oestrogens suppress B virus mRNA levels.	[47,48]
Alcoholic liver disease	Higher alcohol dehydrogenase concentration; Higher volume distribution; higher access to treatment; Lower influence of genetic and epigenetic factors.	Lower alcohol dehydrogenase concentration; Lower volume distribution; lower access to treatment; Higher influence of genetic and epigenetic factors.	[38,39,40]
Autoimmune disease	Lower frequency; Similar access to treatment; Lower influence of genetic and epigenetic factors.	Higher frequency; Similar access to treatment; Higher influence of genetic and epigenetic factors.	[49,50,51,52]
NAFLD	Higher visceral fat.	Protective role of estrogens.	[53,54,55]

**Table 2 jpm-13-00558-t002:** Sex- and gender-related differences and management in T2D and CLD.

	T2D		CLD		References
	Male	Female	Male	Female	
Diet/Lifestyle	Low adherence to diet	Healthier than men, higher attention to diet.	Regardless of gender, some caution needs to be taken for cachexia and sarcopenia in cirrhotic patients.		[98,104]
Statins	No gender differences.	No gender differences.	Contraindicated in the data sheet of drugs for cirrhotic patients. Possible positive effect against HCC.		[105,106]
Ezetimibe	No gender differences.	No gender differences.	Safe in NAFLD/NASH; reduction of AT and GGT levels, lower steatosis and hepatocyte ballooning. No effects on hepatic inflammation and fibrosis. Significant improvement in liver fat when combined with rosuvastatin.		[107,108]
Fibrates		Minor protective effect of fenofibrate on CV events in females.	Pemafibrate significantly improves liver function, serum TG and liver stiffness in NAFLD patients.		[109,110]
Pioglitazone		Greater metabolic benefit (glycaemic control, HOMA IR and lipid improvement). More frequent side effects such as oedema.	Current guidelines indicate Pioglitazone for T2D patients with NASH.		[111,112]
PCSK9i	No gender differences.	Higher circulating concentrations, especially after menopause. Potential regulatory role played by oestrogens.	Correlation between hepatic fat content and PCSK9 levels in NAFLD early stage, not present in the late stages of liver disease. PCSK9 levels are lower in cirrhotic patients and do not correlate with Child–Pugh score, MELD score, bilirubin or AT.		[113,114,115]
Metformin	Greater gastro-intestinal side effects.	Lower doses needed.		Better glycaemic targets reached.	[116,117,118,119]
			Lower doses in cirrhotic patients, especially with an eGFR lower than 30 mL/min/1.73 m^2^.		
DPP-4		Improvement of heart function, blood pressure and body weight.	Caution for hepatotoxicity (Vildagliptin).		[120,121]
GLP-1 RAs	Great effects on Hb1Ac (Exenatide), not present with Dulaglutide.	Greater side effects; higher weight loss.	Great effectiveness for NASH (Liraglutide). Caution in advanced liver disease, limited data about safety and efficacy.		[122,123,124]
SGLT-2i		Greater CV benefits (heart failure).	Improvement of ALT levels, ascites and oedema in decompensated cirrhosis.		[76,125,126]

CLD: chronic liver disease; DPP-4i: dipeptidyl peptidase-4; SGLT-2i: sodium–glucose cotransporter 2 inhibitors; GLP-1 Ras: glucagon-like peptide-1 receptor agonists; PCSK9i: proprotein convertase subtilisin/kexin type 9 inhibitors; eGFR: estimated glomerular filtration rate.

## Data Availability

Not applicable.

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
