# Peer review of "Impact of Sex and Gender on Clinical Management of Patients with Advanced Chronic Liver Disease and Type 2 Diabetes"

_jpm, 2023, doi:10.3390/jpm13030558_

Round 1
Reviewer 1 Report
Dear authors,
The manuscript submitted by Dr. Licata and her colleagues summarize the data regarding the management of chronic liver disease in patients with type 2 diabetes, in relation to sex and gender. Please find my suggestions and comments below:
· Pag. 6: The SGLT2 abbreviation must be explained. Also, the GLP-1 RAS abbreviation must be explained. Please correct GLP1 with GLP-1
· Pag. 10: “… with an increased cardiovascular risk in the woman as compared with that of man.” A reference must be added.
· Pag. 12: The HCC abbreviation should be explained
· Pag. 13: The MELD abbreviation should be explained
· Pag. 15: The CLD abbreviation should be explained
· Table 3: It would be better to insert a column with references. Also, authors must justify the information regarding the differences on HF outcomes of iSGLTs between sexes
· Pag 16: Authors should present information regarding ezetimib and PCSK inhibitors.
· Pag 16: “They are also safe in cirrhotic patients,…” a reference should be added
· Pag. 21: “A meta-analysis of three randomized trials conducted with SGLT-2I… “ a reference should be added
· Pag. 22: “… possibly due to rela-tively greater weight loss due to reduced fat mass with GLP-1RA compared to SGLT-2I.” a reference should be added
· The Conclusions section should be summarized.
Author Response
Dear Reviewer # 1
We thank for the suggestions above our paper entitled: IMPACT OF SEX AND GENDER ON CLINICAL MANAGEMENT OF PATIENTS WITH ADVANCED CHRONIC LIVER DISEASE AND TYPE 2 DIABETES.
- Pag. 6: We have explained The SGLT2 abbreviation, and the GLP-1 RAS abbreviation must be explained. GLP1 has been corrected with GLP-1
- Pag. 10: “… with an increased cardiovascular risk in the woman as compared with that of man.” A reference must be added.
We added: Shin JA, Lee JH, Lim SY, Ha HS, Kwon HS, Park YM, Lee WC, Kang MI, Yim HW, Yoon KH, Son HY. Metabolic syndrome as a predictor of type 2 diabetes, and its clinical interpretations and usefulness. J Diabetes Investig. 2013 Jul 8;4(4):334-43. doi: 10.1111/jdi.12075. Epub 2013 May 28
- Pag. 12: The HCC abbreviation have been explained
- Pag. 13: The MELD abbreviation have been explained
- Pag. 15: The CLD abbreviation have been explained
- Table 3: A column with references has been inserted.
- Pag 16: Information regarding ezetimibe and PCSK inhibitors have been added.
- Pag 16: “They are also safe in cirrhotic patients,…” a reference should be
We added: Kaplan DE, Serper MA, Mehta R, et al. Effects of hypercholesterolemia and statin exposure on survival in a large national cohort of patients with cirrhosis. Gastroenterology 2019; 156: 1693- 1706.e12.
- Pag. 21: “A meta-analysis of three randomized trials conducted with SGLT-2I… “ a reference should be
We added: Staplin N, Roddick AJ, Emberson J, Reith C, Riding A, Wonnacott A, Kuverji A, Bhandari S, Baigent C, Haynes R, Herrington WG. Net effects of sodium-glucose co-transporter-2 inhibition in different patient groups: a meta-analysis of large placebo-controlled randomized trials. EClinicalMedicine. 2021 Oct 26;41:101163. doi: 10.1016/j.eclinm.2021.101163
- Pag. 22: “… possibly due to relatively greater weight loss due to reduced fat mass with GLP-1RA compared to SGLT-2I.”
We added: Gurgle HE, White K, McAdam-Marx C. SGLT2 inhibitors or GLP-1 receptor agonists as second-line therapy in type 2 diabetes: patient selection and perspectives. Vasc Health Risk Manag. 2016 Jun 4;12:239-49. doi: 10.2147/VHRM.S83088.
- The Conclusions section have been summarized.
Reviewer 2 Report
Dear editors
thanks for invitation to review this manuscript.
Actually, I have read the review and I found no rules for systematic review were followed and tables and conclusions are not referred to any evidence rather than free readings.
I advise authors to follow the rules for systematic reviews or meta analysis to get an evidence based conclusions
Author Response
Dear Reviewer # 2
Thank you for reviewing our work.
Considering the huge amount of data present in the literature regarding type 2 diabetes, advanced chronic liver diseases of various etiologies, as well as the gender studies that are emerging in recent years, we thought to provide the reader with a narrative review that could provide a clinical help to the clinician who is managing the patient with type 2 diabetes and advanced chronic liver disease.
We agree that systematic review and/or meta-analysis are adequate scientific tools to review a complex topic and reach a higher level of quality, but the purpose of this narrative review meanwhile was to frame the problem of diabetes therapy in patients with chronic liver disease who come to internal medicine clinics every day, who usually do not have a single pathology, diabetes, or cirrhosis, but often these coexist with others. Furthermore, the fact that now many studies consider the emerging role of sex and gender issue in relation to therapeutic response, efficacy, and safety, would have forced us to face not one, but at least 3 systematic reviews, one for each of the topics analyzed.
However, thanks to your valuable comments, we think that doing a major revision, as the journal requested, we have considerably improved our work.
Reviewer 3 Report
The manuscript submitted by the authors does not have enough form and quality characteristics to be considered for publication in its current state. Here are some reasons:
· It has a significant number of references that could be cited systematically, many of them supporting almost the same ideas. Moreover, important references on the field were not included (e.g.).
· From the title onwards, the authors propose that "sex" and "gender" are independent/synergistic factors of chronic liver disease and type 2 diabetes. Although throughout the manuscript the authors attempt to establish/highlight the so-called " sex- and gender difference" (First sentence in the introduction), this difference is not clear (a fact previously discussed by others. Doi: 10.3389/fendo.2018.00220), partly because in the search strategy the authors surely used these terms as "independent keywords" and, therefore, in the evidence gathered these terms are overlapped, confused, or considered synonymous (in clinical practice although not humanistic). The authors must explain from the beginning the differences of these terms, from an internal medicine point of view (not humanistic) and how this difference is radical for individual (personalized) and public health (it is recommended to see Doi: 10.1007/978- 3-642-30726-3_1, 10.1016/S0140-6736(20)31561-0, 10.1186/s41073-019-0068-4).
· Even though the authors reviewed a significant number of references to build their manuscript, it is presented in a narrative (and even, on some occasions, anecdotal) and non-systematic way. This substantially decreases the uniqueness of the manuscript compared to other systematic reviews and meta-analyses, including some cited in the manuscript (Doi:10.1016/S0140-6736(14)60040-4, 10.3389/fendo.2018.00220). One or two figures and systematic review tables are strongly suggested.
· Authors must strictly follow the journal guidelines for formatting the manuscript.
Author Response
Dear Reviewer # 3
Thank you for reviewing our work.
Considering the huge amount of data present in the literature regarding type 2 diabetes, advanced chronic liver diseases of various etiologies, as well as the gender studies that are emerging in recent years, we thought to provide the reader with a narrative review that could provide a clinical help to the clinician who is managing the patient with type 2 diabetes and advanced chronic liver disease.
We agree that systematic review and/or meta-analysis are adequate scientific tools to review a complex topic and reach a higher level of quality, but the purpose of this narrative review meanwhile was to frame the problem of diabetes therapy in patients with chronic liver disease who come to internal medicine clinics every day, who usually do not have a single pathology, diabetes, or cirrhosis, but often these coexist with others. Furthermore, the fact that now many studies consider the emerging role of sex and gender issue in relation to therapeutic response, efficacy, and safety, would have forced us to face not one, but at least 3 systematic reviews, one for each of the topics analyzed.
However, thanks to your valuable comments, we think that doing a major revision, as the journal requested, we have considerably improved our work,
- modifying the introduction better explaining the differences between sex and gender according one of the suggested references, (see at references list, #1. Mauvais-Jarvis F et al, Lancet 2020 ),
- reducing the multiple sometimes repetitive references.
- Further, to make more interesting our work, we also add within the therapy section a paragraph on PCSK9 inhibitors (as request by Reviewer # 1), one on Fibrates, fenofibrate and pemafibrate, and another on Pioglitazone.
- Again, we have eliminated a table that could cause confusion.
- we have adjusted tables 2 and 3 instead, adequately supplying them with references by each topic.
- lastly, we made a figure, according to the suggestion of the Reviewer # 4.
- The journal guidelines for formatting the manuscript have been followed
- The English have been revised by a native speaker.
Reviewer 4 Report
Dear authors,
After the review process, I have several comments: the paper has a major lack of data regarding microbiota bioactivity; the authors should add data related to a comparative fingerprinting of the human microbiota in diabetes and cardiovascular diseases; if you mention personalized diagnostic and therapeutic should take into considerations data related to the supplements that have as a target T2D, for example; you should include a figure that explains the following data: correlations between microbiota bioactivity and bioavailability of functional compounds, that act as a support in T2D therapy.
Best regards!
Author Response
Dear Reviewer# 4
Thanks for your useful and clever suggestion.
To make this work more complete, we wanted to insert a paragraph on the influence that sex and gender have on the gut microbiota of patients with NAFLD and type 2 diabetes.
We have also included an explanatory figure in this section (figure 1), with a caption showing the type of diet and the bioactive nutritional compounds, polyphenols and flavonoids contained in the Mediterranean Diet, and the positive effect they have in promoting eubiosis of the gut; on the contrary it is also explained how the negative effect of the Western diet favors intestinal dysbiosis, favoring liver inflammation, pro inflammatory cytokines patterns and increasing of ROS specie.
We have also added the requested reference contextually: Basista Rabina Sharma, Swarna Jaiswal, P.V. Ravindra. Modulation of gut microbiota by bioactive compounds for prevention and management of type 2 diabetes. Biomedicine & Pharmacotherapy,152, 2022, 113148, doi.org/10.1016/j.biopha.2022.113
Round 2
Reviewer 3 Report
Thanks for having accepted most of my suggestions
Reviewer 4 Report
No other comments.